# Detection of Cardiovascular CRP Protein Biomarker Using a Novel Nanofibrous Substrate

**DOI:** 10.3390/bios10060072

**Published:** 2020-06-24

**Authors:** Isaac Macwan, Ashish Aphale, Prathamesh Bhagvath, Shalini Prasad, Prabir Patra

**Affiliations:** 1Department of Electrical and Bioengineering, Fairfield University, Fairfield, CT 06824, USA; 2Department of Biomedical Engineering, University of Bridgeport, Bridgeport, CT 06604, USA; aaphale@my.bridgeport.edu (A.A.); ppatra@bridgeport.edu (P.P.); 3Department of Biomedical Engineering, Saint Louis University, St. Louis, MO 63103, USA; pbhagvath@slu.edu; 4Department of Bioengineering, University of Texas at Dallas, Richardson, TX 75080, USA; shalini.prasad@utdallas.edu; 5Department of Mechanical Engineering, University of Bridgeport, Bridgeport, CT 06604, USA

**Keywords:** biosensors, C-reactive protein, carbon nanotubes, electrospinning, electrochemical impedance

## Abstract

It is known that different diseases have characteristic biomarkers that are secreted very early on, even before the symptoms have developed. Before any kind of therapeutic approach can be used, it is necessary that such biomarkers be detected at a minimum concentration in the bodily fluids. Here, we report the fabrication of an interdigitated sensing device integrated with polyvinyl alcohol (PVA) nanofibers and carbon nanotubes (CNT) for the detection of an inflammatory biomarker, C-reactive protein (CRP). The limit of detection (LOD) was achieved in a range of 100 ng mL^−1^ and 1 fg mL^−1^ in both phosphate buffered saline (PBS) and human serum (hs). Furthermore, a significant change in the electrochemical impedance from 45% to 70% (hs) and 38% to 60% (PBS) over the loading range of CRP was achieved. The finite element analysis indicates that a non-redox charge transduction at the solid/liquid interface on the electrode surface is responsible for the enhanced sensitivity. Furthermore, the fabricated biosensor consists of a large electro-active surface area, along with better charge transfer characteristics that enabled improved specific binding with CRP. This was determined both experimentally and from the simulated electrochemical impedance of the PVA nanofiber patterned gold electrode.

## 1. Introduction

Detection of biomarkers specific to a particular disease is a standard method for diagnosis and can be used for successful detection of various diseases [1]. A number of proteins considered as biomarkers are secreted at an early onset of a disease when the symptoms are yet to develop. A typical source to detect such biomarkers can be a patient’s blood sample. C-reactive protein (CRP) has long been used as a marker of systemic inflammation, where the level of CRP increases several hundred fold within hours of an inflammatory occurrence [2] and it is found to be over-expressed in heart disease [3]. Previous research has shown that CRP is dominant in the instigation of several pathogenic pathways that may cause atherosclerosis, a precursor to cardiovascular disease [4]. According to the Center for Disease Control and Prevention, heart attack is the leading cause of death in the US with an estimated 785,000 Americans having a new coronary attack and approximately 470,000 with a recurrent attack [5]. Thus a rapid and ultra-sensitive detection of CRP from a patient’s serum sample may be clinically significant in diagnosing cardiac events such as acute myocardial infarction. In recent years, the trend in healthcare has been towards developing devices that support personalized medicine with benefits such as faster diagnosis and therapeutic turnaround time. 

Nanotechnology offers a wide range of materials that can be used both as biological recognition elements and transducers for a typical biosensor with exceptional levels of sensitivity that can be used to monitor biomarkers from different diseases. Novel properties of nanoscale materials, especially in the form of nanofibers, such as larger surface to volume ratios and size-based confinement of biomolecules, give them the potential to increase the sensitivity of the diagnostic devices by several orders of magnitude [6,7,8,9,10,11]. This property can be used to detect protein biomarkers such as CRP at very low concentrations and also give an early indication of a heart condition [12,13].

Some of the previous efforts in developing biosensors based on nanofibers involved citric acid-decorated nylon nanofibers for the detection of 3-phenoxybenzoic acid (3-PBA), a common human urinary metabolite [14], and polyvinyl alcohol-co-ethylene (PVA-co-PE)-based nanofibrous membranes for the detection of chloramphenicol (CAP) residues in milk [15]. One of the aspects of polyvinyl alcohol (PVA) and carbon nanotube (CNT)-based nanofibrous biological recognition elements is that they offer spatial confinement similar to the intracellular environment through porosity. The binding affinities and the rate at which a typical self-assembly process takes place can benefit from several physical constraints during the biosensing of macromolecules in vitro [16,17,18]. The nanotexturing of the sensor platform enables minimization of excluded volume of the biomolecules not participating in the interaction with the biomarker, which is also evident from the increase in the concentration that is analyzed through finite element analysis of diluted species [19]. Furthermore, even though individual fibers may have been laid on the electrode, there are still multiple fibers exhibiting a certain thickness similar to a membrane-like matrix but taking advantage of a higher specific surface area.

One of the major challenges faced by nanomaterial-based biosensors is the sensitivity and reproducibility of the results, which arises mainly because of the extreme difficulty in detecting a small quantity of protein biomarkers in serum samples [20,21,22]. Hence, determining the concentration of trace biomarkers in a complex mixture is a challenge in patient diagnosis. 

Amongst several diagnostic biosensors, most of the conventional ones rely on fluorescent labeling or dyes for gathering the data [23]. Various sensing techniques that are applied to detect biomolecules at low concentrations are electrochemical analysis, surface plasmon resonance, electromagnetic measurements or mechanical actuation. Electrochemical sensors allow a label free detection of biomolecules by detecting and measuring the electrical signal [24,25,26]. A variety of electrochemical techniques based on integrated devices have been employed for label free and ultra-sensitive detection of different biomarkers. These methods are based on the principles of charge transfer [27,28,29], radiofrequency [30], complementary metal oxide semiconductors (CMOS) [31], capacitance [32], or impedance [33]. CMOS-based sensors are limited by the complexity of the fabrication technique, which leads to an extremely high cost of the overall biosensor device. In contrast, impedance and capacitance-based measurements are potential techniques for sensing a variety of biomolecules, mainly due to the low power consumption, ease of miniaturization and relatively low cost [34]. It is known that the performance of any biosensor depends on the immobilization of biomolecules on a biocompatible electrode surface. Therefore, the specificity of a biosensor can be increased by increasing the overall specific binding sites on the surface of the electrode. Hence, a biocompatible electrode, with an extremely high surface area that can engender spatial confinement for favorable binding events, would be a preferred platform for antigen–antibody interactions to achieve high sensitivity and specificity.

Here, we report, an extremely sensitive diagnosing platform with an ability to detect CRP concentrations up to fg mL^−1^ in both phosphate buffer saline (PBS) and human serum (hs). A microchip with seven interdigitated micro-comb capacitors is integrated with a randomly aligned electrospun PVA nanofibrous mesh. Owing to the random alignment of the nanofibers, the mesh showed high porosity for macromolecule confinement and an extremely small individual fiber diameter (~250 nm), along with a larger surface area to volume ratio, providing efficient antigen–antibody interactions. Furthermore, the signal transduction is found to be chemo-electro ionic, with the protein binding resulting in a modulation of the electrical double layer at the nanofibrous surface. A 45% change in the measured impedance from the antibody saturation baseline was observed while detecting the lowest detectable dose of 1 fg mL^−1^. Finally, a change in the oxidation current for a gold substrate with and without PVA/CNT/anti-CRP functionalization provides crucial insights on the charge transfer resistance at the double layer capacitance interface between the metal electrode and the bulk electrolyte. The results obtained through a finite element model are in close agreement with the experimental data.

## 2. Materials and Methods

The nanoweb was fabricated using an electrospinning process (Figure 1a). The electrospinning set up comprises a high voltage power supply and a dual syringe pump unit. A positive electrode from the high voltage power supply is connected to the needle. The nanoweb is collected on a grounded metal collector covered with insulating polyethylene. To produce the PVA-CNT nanocomposite fibrous nanoweb, we applied 15 kV of voltage across 10 cm distance between the collector and the needle tip with a flow rate of the CNT dispersed polymer solution of 1.5 mL/h.

The nanoweb biosensor chip was comprised of a base printed circuit platform. The platform is overlaid with the nanoporous nanoweb layer, which is then encapsulated by microfluidic manifolds manufactured using polydimethylsiloxane for fluid encapsulation and confinement. Based on the SEM images, the area of the pores is estimated to be in the range of 100 to 500 square nanometers. CRP is used in phosphate buffered saline as well as 50% human serum for evaluating the sensor performance.

Printed circuit board (PCB) chips comprising gold comb shaped designs of dimensions 10 mm × 10 mm in length; a finger width of 1 mm with spacings of 1 mm were manufactured with a FR4 passivation layer. After cleaning the surface of these chips with 10 mL of isopropyl alcohol (IPA) and air drying the samples for 10 min, nanowebs constituting 10 mg of CNT in 10% PVA were electrospun (Figure 1b). Figure 1c shows a blown-up image of the individual fingers and the presence of the nanoweb in between.

To ensure that the nanofibers are deposited on the gold microelectrodes and not on the FR4 (glass reinforced epoxy laminate printed circuit board) surface of the microchip itself, both the ‘Working’ and the ‘Reference’ electrodes were also grounded. Interestingly, the above mentioned procedure results in selective deposition of nanocomposite fibers only on the conductive electrodes allowing the FR4 surface to be clean without any deposition. The average diameter of the nanocomposite fibers is ~250 nm, as seen from the SEM image in Figure 1d. Based on the flow rate of 1.5 mL/h and the electrospinning time of ~90 min, the thickness of the electrospun nanofibrous substrate was found to be ~0.5 mm. The presence of CNTs in the polymer matrix helps to stretch the nanofibers under the electric field, enabling continuous stretching and thereby forming a uniform fiber diameter without beaded morphology owing to the conductive nature of the CNTs. This ensures uniform fiber distribution, creating a spatially confined environment for better antigen–antibody interactions.

The dimensions of the nanowebs used for the experiments were 13.2 mm × 13.2 mm. The nanowebs were overlaid onto the metallic sensor surface using tweezers. The nanoweb was encapsulated with a polydimethylsiloxane (PDMS) manifold with dimensions of 13 mm × 13 mm. The manifold has a groove of 5 mm depth to enable the localization of the nanoweb onto the sensor surface. All the steps pertaining to the detection assay were performed on the assembled sensor chip.

Detection of the protein binding event was achieved using electrochemical impedance spectroscopy (EIS), and electrical double layer (EDL) capacitance measurement [9,10,12]. Briefly, the EDL comprises two components: the solution resistance (R_s_) and the double layer capacitance (C_dl_). The binding of the protein biomolecules to the covalently anchored antibodies in the nanotextured surface resulted in a modulation to the charge at the electrical double layer formed at the solid/liquid interface [12]. This change in charge produced a change in C_dl_. The capacitance change was measured as an impedance change using an impedance analyzer (Impedance/Gain Phase Analyzer, Autolabs, Avon, IN, USA).

Oscillating AC fields of 0.05 V were applied, and the frequency was scanned over a range of 40 Hz–10 kHz. Frequencies only of the lower orders (up to 10 kHz) were considered for this study. At these low frequencies C_dl_ undergoes major variations during protein binding at the electrical interface. The sample volume was maintained throughout the experimental trials at 150 µL, sufficient to completely wet the sensor surface. This kept the solution resistance R_s_ constant within this frequency regime [10]. This EIS technique produced measurements that were purely the result of the changes in C_dl_, and indicated protein binding.

Two sets of dose response experiments were performed to test the capabilities of the nanoweb sensor platform using electrochemical impedance measurements. The first set was focused on detecting CRP when aliquoted in isotonic buffer solution, 0.15 M phosphate buffer saline (PBS). Aliquots of hs-CRP ranging from 1 fg/mL to 100 ng/mL were prepared on a logarithmic scale. These concentrations were selected because they represent clinically relevant concentrations of CRP in physiological conditions. It is known that dithiobis succinimidyl propionate (DSP), which is a covalent linker, has a thiol (sulphur) group at one end that has high affinity for the gold surface while the other end, which is the NHS group, binds with the biological macromolecule (anti-CRP in this case) [35,36]. Based on the highly porous structure of the nanofibrous substrate, the DSP linker would permeate through these pores and anchor on the underlying gold substrate. This unique substrate now allows the interaction of the analyte (CRP) with the anti-CRP within these pores, increasing the overall sensitivity of the biosensor. Thus, after integrating the nanoweb mat and PDMS manifold onto the chip, 150 μL of 10 mM DSP crosslinker was injected into the manifold and incubated for 30 min at room temperature. The sensor surface was subsequently washed three times with 0.15 M PBS and baseline PBS measurements were taken. Following the crosslinker deposition, 150 μL of 50 μg/mL anti-CRP was incubated onto the sensor surface at 4 °C for 2 h to immobilize the receptors. After immobilizing anti-CRP on the sensing surface, 0.15 M PBS wash was performed three times followed by incubation of super block, a blocking protein that reduces the non-specific binding, on the sensor surface for 30 min at room temperature to minimize non-specific binding, which was again followed by a 0.15 M PBS wash that was performed three times. A zero dose, corresponding to 0.15 M PBS, was injected into the manifold and the measured impedance was considered as the baseline. All the impedance measurements for different dose concentrations of CRP were normalized to this baseline measurement. Starting from the lowest dose within the range, 150 μL of CRP spiked buffer was injected into the manifold, incubated for 15 min and impedance measurement was taken. The change in impedance from baseline measurement was calculated and converted into percentage change of impedance from baseline readings.

The second set of experiments focused on detecting CRP when aliquoted in CRP free human serum (hs). Aliquots of hs-CRP ranging from 1 fg/mL to 100 ng/mL were prepared on a logarithmic scale. After immobilization of anti-CRP onto the sensor surface, followed by blocking and washing, the zero dose, corresponding to CRP-free human serum, was injected into the manifold. EIS measurements were taken after 15 min incubation and considered as the baseline measurement. CRP-spiked human sera of different concentrations were subsequently injected onto the sensor surface and measured impedance was converted to percentage change from baseline impedance.

For finite element analysis, a portion of the microelectrode sensor was taken and electrochemical impedance analysis was carried out to obtain the Nyquist plots. First the area and the volume of the gold surface and the number of gold atoms at the surface of the electrode were calculated by using the concentration of redox couple as 10^−3^ mol/m^3^. At the initial stage, only oxidizing agents are present at the electrode surface and, since the electrode is bare, these agents easily get electrons from the gold surface and get reduced. The initial reaction rate, K_0_, at this stage was found to be ~2.5 × 10^−4^ cm/s. Utilizing the transport of diluted species module, DSP was diffused on the gold electrode and the surface concentration of DSP was calculated after 1600 s. From the concentration value obtained, it was found that the DSP monolayer occupied ~30% of the electrode surface with each molecule shielding around 8 gold atoms. According to this insight, ~50% of the gold atoms at the electrode surface are shielded, which affects the forward reaction rate coefficient, K_f_, of the redox reaction and in turn reduces the K_0_ by another 50%, making it ~1.25 × 10^−4^ cm/s. In the same manner, the concentrations of CRP and anti-CRP were also calculated. Considering the size of anti-CRP and CRP, it was found that they could shield around 22 and 62 gold atoms respectively. Thus, 60% of the gold atoms would be shielded by anti-CRP and 70% by CRP reducing the K_0_ by 40% (1 × 10^−4^ cm/s of the initial value) after anti-CRP deposition and by 30% (7.5 × 10^−5^ cm/s) after CRP deposition. In the case of the nanofiber meshed electrode, we can see from the design that some part of the gold surface is occupied by nanofibers and it was estimated to be covering ~55% of the gold atoms. Thus, the initial value for K_0_ was found to be ~1.125 × 10^−4^ cm/s. Taking the same assumption for the number of gold atoms shielded by DSP, anti-CRP and CRP, we get the following values of K_0_: 0.5625 × 10^−4^ cm/s after DSP deposition, 0.45 × 10^−4^ cm/s and 0.3375 × 10^−4^ cm/s after anti-CRP and CRP deposition, respectively.

## 3. Results

### 3.1. Determination of CRP Using EIS

The binding of the protein biomolecules to the covalently anchored antibodies on the nanotextured surface results in the modulation of charge at the electrical double layer (C_dl_), as seen in Figure 2. The binding of CRP to its antibody at this interface produces a specific and measurable change of impedance across the electrode. As the binding of the biomolecules occur directly on the sensor surface and is not mediated through a redox probe, the impedance changes are non-inductive in nature.

The charge transfer resistance (R_ct_) experienced by the leakage charge through EDL at the sensor/analyte interface and the Warburg impedance (Z_w_), which is the diffusional impedance experienced by ions in bulk buffer, are not dominant at lower frequencies and mostly treated as constant [33]. The solution resistance (R_s_) is also considered constant as the analyte volume is kept constant throughout the experiments. Therefore, impedance measured at the lower frequencies, i.e., below 1 kHz, are representative of the changes to the EDL due to biomolecule binding. Hence, impedance data was represented for 100 Hz, which is a representative frequency of the low frequency range. A higher quantity of target protein in the analyte produces a higher amount of shift in the impedance from baseline and thereby allows for quantitative detection of the protein.

A biosensor dose response analysis was performed based on the two sets of experiments with the CRP in PBS buffer and human serum (hs) to test the capabilities of the nanofibrous sensor platform as shown in Figure 3.

As can be seen from Figure 3a, the synthesized nanofibrous sensor platform was able to detect CRP from 100 ng/mL down to 1 fg/mL, indicating a unique ability of this platform to detect a minimum concentration of CRP. Each concentration was tested on different electrodes to avoid the removal of residual protein and successive regeneration of the sensor surface after every test. Also from Figure 3b, comparing the impedance values across this regime between the PBS and hs, it is found that there is relatively larger change in impedance for hs from ~360 ohms to ~220 ohms compared to PBS (~220 ohms to ~140 ohms), showing that this technique has a lot of potential to detect biological macromolecules in their native environment.

### 3.2. Biosensor Performance

The sensor performance was determined by the impedance change from the antibody saturation baseline expressed as a percentage change (Figure 4a,b). As the dose of CRP in the solution increased, the corresponding measured absolute impedance value from the biosensor decreased. Figure 4a shows the percentage change in the impedance with respect to the various doses of CRP in PBS and hs. The limit of detection (LOD) in this case was estimated to be 1 fg mL^−1^ with a percent change of 38% from the antibody baseline impedance, which is the anti-CRP mounted on the PVA/CNT-coated electrode control. The sensitivity of the sensor was computed by determining the dose of the antigen that gave at least 10% change from the baseline signal, where 10% change correlated to the signal background. Figure 4a also shows the dose response of CRP in 50% human serum. Initially the background signal from the antibody saturated sensor was determined by injecting 50% human serum directly on the sensor surface. The assay on the nanofibers was constructed in a similar manner for the detection of CRP from PBS samples. We observed a 45% change in the case of hs and a 38% change in the case of PBS for the impedance while detecting the lowest dose of 1 fg mL^−1^. The percentage change in impedance ranged from 45% to 70% (hs) and 38% to 60% (PBS) over a concentration range of 1 fg mL^−1^ to 100 ng mL^−1^. Saturation in measurement was observed for the CRP dose of 10 pg mL^−1^, indicating a limit on linearity from the performance standpoint.

### 3.3. Cross-Reactivity Test

Cross-reactivity tests were performed to determine the selectivity of the biosensor using anti-troponin-T as the antibody and CRP as the antigen. The sensor was immobilized with 1 μg mL^−1^ of anti-troponin-T and was prepared in a manner described earlier for the case of anti-CRP. The cross reactivity was studied by immobilizing anti-troponin-T with CRP in PBS and was compared to the performance of chips with anti-CRP and CRP in both hs and PBS (Figure 5). Dose response studies were conducted in a similar manner to those described in the previous subsection. As before, the impedance response for CRP interaction with anti-CRP ranged from ~38% to 60% in PBS and ~45% to 70% in hs. On the other hand, there was a very low response of the CRP interaction with anti-troponin-T, with the percentage change in impedance below 10%. These results indicated the robustness as well as the selectivity of the designed sensor and its response.

### 3.4. Finite Element Analysis to Determine the Change in Impedance on Detection of CRP

Both the electrodes, bare gold and the one covered with nanofiber mesh, were simulated for the change in impedance when depositing DSP, anti-CRP and CRP and were compared with the controls (absence of these molecules). Figure 6a shows the FEA model that was constructed with the same dimensions as the real electrode. Three layers of randomly oriented nanofibers were modeled on the constructed electrode having the same diameter (~250 nm) as the real nanofibers. As can be seen from Figure 6b,c, the Nyquist plots were obtained for both of these scenarios and the change in impedance was compared. It can be seen that a significant change in impedance with deposition of each layer on the bare gold electrode as well as on the nanofiber meshed gold surface was obtained. In addition, it is found that the impedance due to the CRP deposition on bare gold electrodes is less compared to the one with a nanofiber mesh in agreement with the real-time impedance data that was obtained showing the beneficial effects of the nanofiber mesh owing to its porosity.

## 4. Discussion

The PVA/CNT nanofiber-embedded biosensor device exhibits an amplification of the measured electrochemical impedance signal associated with the protein binding. CRP immobilization is achieved using a standard immunoassay protocol. The transduction is chemo-electro ionic with the protein binding resulting in a modulation of the electrical double layer at the interface of the nanofibers. Charged groups are present both on the hydrophilic surface as well as on the hydrophobic residues that are interior to the protein molecule [37]. Upon the application of voltage, the charged surface facilitates either repulsive or attractive forces on the ions at the electrode interface leading to a change in the dielectric thickness across the electrode [34]. These changes in the dielectric thickness are directly proportional to the thickness of the electrical double layer. When the electrode surface becomes nanotextured as in this particular study, the effect of the modulation on the dielectric thickness is amplified primarily due to the segmentation of the electrical double layer.

The changes in the impedance were comparable in both the ionic buffer as well as in human serum. The minimization of high background in the human serum buffer may be attributed to the sieve-like behavior of the nanofibers that provide size matched confinement of the target proteins. Furthermore, given the nature of the substrate, being nanofibrous and porous, it turns out to be a better microenvironment for the analyte to interact. In addition, the pores within the nanofibrous substrate would provide an environment that may promote the phenomenon of macromolecular crowding that typically exists in confined spaces, thereby altering the behavior of human serum [16,17]. Interestingly, it was observed that the protein association is significantly enhanced at lower concentrations, which in turn enhanced the sensitivity of the biosensor device.

The dimensions of the composite nanofibers are ~100 µm in length and ~250 nm in diameter. The nanofibers make a good biochemical transducer due to the presence of CNT and are highly suitable for achieving electrical signal amplification associated with the detection of the biomolecules. The resulting nanostructure comprises a non-periodic array of nanoscale confined spaces, which are electrically connected through the metallic micro electrode. The nanofibers have been surface functionalized with a protocol similar to the enzyme linked immunosorbent assay.

The most important characteristic of a biosensor is the calibration dependence of the impedance signal (Z) on the concentration of the analyte (C) [38]. The behavior of the calibration curve can be understood using a suitable mathematical equation such as the Hill isotherm model [39], which is used to describe the binding of different species onto homogenous substrates to fit the experimental data. According to this model, the antigen binding ability at one site on the macromolecule may influence different binding sites on the same macromolecule and the adsorption process is a cooperative phenomenon. As seen in Figure 4, the biosensor response in the form of percent change in impedance (%Δ Z) and difference impedance (ΔZ), are closely fitted with the following Hill formula:(1)y=Vmax 
where y is the biosensor response (signal Z), Vmax is the maximum state of the reaction reached at concentration  x, k  is the antigen concentration that binds the receptor sites at half concentration and n is the Hill coefficient.

For Figure 4, the values of n (0.2221 ± 0.00374 (PBS) and 0.1050 ± 0.0862 (hs)) are calculated based on Equation (1). This low value of Hill’s coefficient at all the concentrations is a measure of the variable free energy of the interactions between biomolecules and surfaces [40,41,42]. In both PBS and hs, Hill’s slope is less than 1, which is an indication of negative cooperative binding, which means that although many binding sites may be present, only specific binding occurs at a given point of time [43]. This observation is also supported by the cross-reactivity study, where at 1 fg mL^−1^ concentration of CRP, the percentage change in the impedance is less than 10%. The values of k are 0.1744 ± 0.0089 fg mL^−1^ (PBS) and 0.0708 ± 0.2357 fg mL^−1^ (hs), respectively.

## 5. Conclusions

A novel biosensor device is fabricated and tested for the detection of CRP protein. This affinity-based biosensor, working on the principle of electrochemical impedance spectroscopy, demonstrated an excellent signal to noise ratio with a detection limit of 1 fg mL^−1^. Furthermore, a 45% (hs) and 38% (PBS) change in the electrical impedance was observed while measuring the lowest detectable dose of 1 fg mL^−1^. It is proposed that within the confinement of the pores of the nanofibrous mesh, the overall specific binding increases, thereby reducing the noise set by non-specific interactions. The signal transduction is chemo-electro ionic with the protein binding resulting in a modulation of the electrical double layer capacitance at the nanofibrous interface. Furthermore, the simulated behavior of the biosensor electrode in the presence and absence of the nanofiber mesh clearly demonstrated a larger change in impedance owing to the nanofiber mesh indicating the role of porosity in the detection of CRP.

## Figures and Tables

**Figure 1 biosensors-10-00072-f001:**
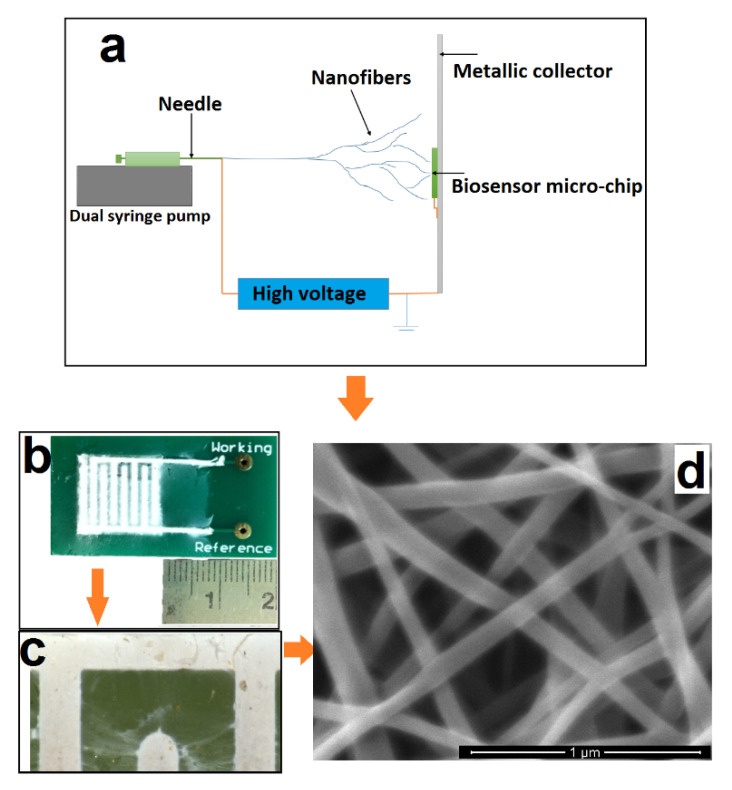
Fabrication of PVA/CNT nanofibers and their deposition on the biosensor: (**a**) Schematic of nanofiber deposition on the biosensor chip using electrospinning; (**b**) Optical image of the actual biosensor with selectively deposited nanofibers; (**c**) Scanning electron microscope (SEM) image of electrospun PVA/CNT nanofibers; (**d**) High resolution SEM of the deposited nanofibers clearly exhibiting the porous nature of the electrospun nanofibers with a diameter of ~250 nm.

**Figure 2 biosensors-10-00072-f002:**
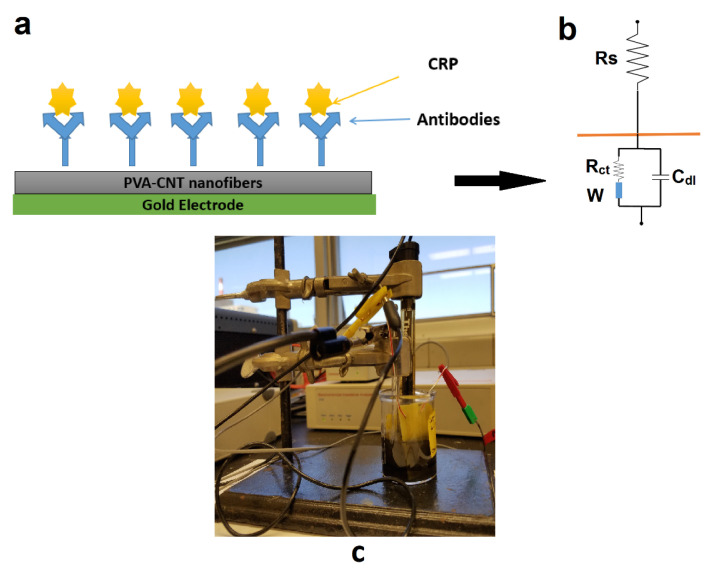
General scheme and electrochemical impedance measurement setup: (**a**) Immobilization of antibody/antigen on the electrospun nanostructured mesh selectively deposited on the interdigitated micro comb gold electrodes; (**b**) Resulting electrical circuit at the electrode/electrolyte interface; (**c**) Experimental setup for the quantification of the electrochemical impedance of the PVA/CNT-coated electrode.

**Figure 3 biosensors-10-00072-f003:**
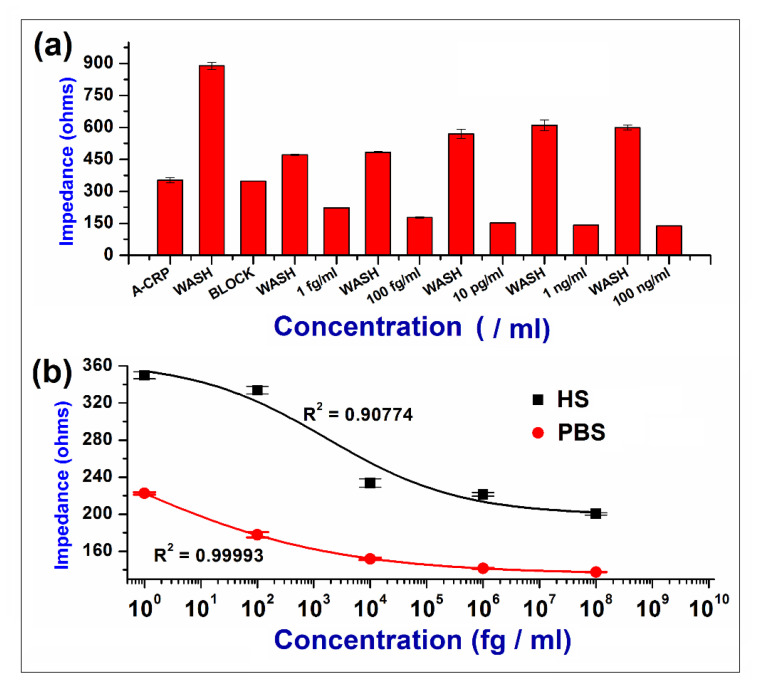
Dose response in PBS and hs: (**a**) Absolute impedance values after each step of antibody immobilization (A-CRP), followed by washing and superblock (Block) deposition to test the sensitivity of the biosensor on individual electrodes; (**b**) Changes in the impedance with respect to the concentration of CRP in both PBS and hs. All the impedance measurements are with respect to the concentration of antigen (CRP) per mL of either buffer (PBS/hs).

**Figure 4 biosensors-10-00072-f004:**
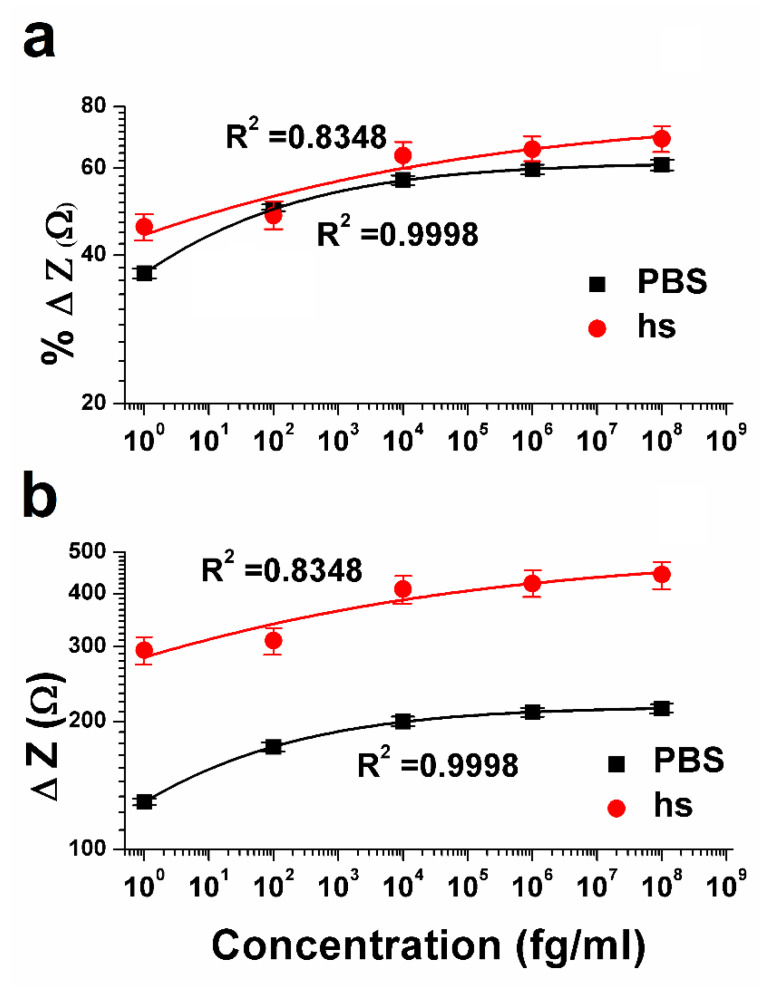
Dose response of CRP in PBS and hs: (**a**) Percentage change in impedance (%ΔZ) corresponding to the concentration of CRP; (**b**) CRP dose dependent change in the impedance (ΔZ).

**Figure 5 biosensors-10-00072-f005:**
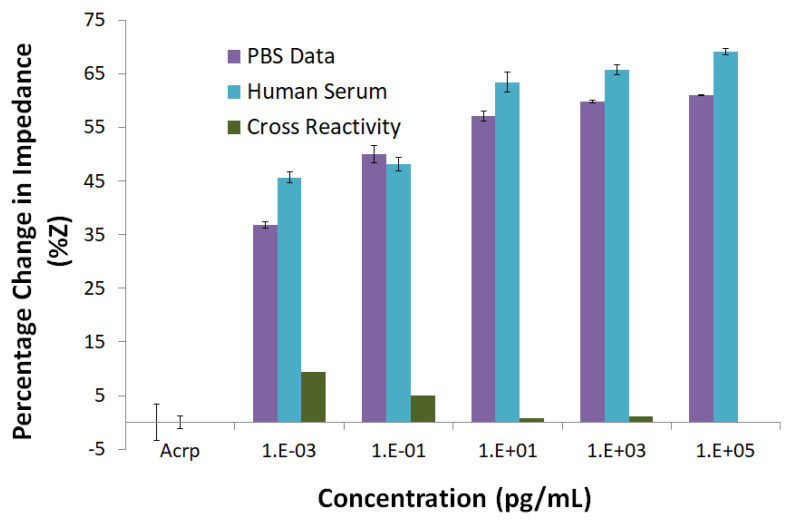
Cross reactivity study. Comparison of CRP binding with anti-CRP in both hs and PBS and cross reactivity of CRP with anti-troponin-T.

**Figure 6 biosensors-10-00072-f006:**
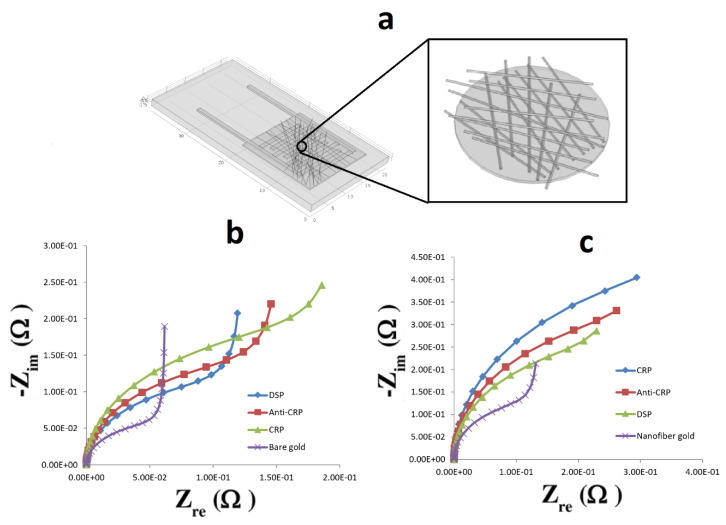
Finite element analysis of the change in impedance due to the presence of CRP, anti-CRP and DSP: (**a**) Modeled gold coated copper electrode on FR4 material; (**b**) Change in impedance due to the presence of DSP, anti-CRP and CRP on bare electrodes; (**c**) Change in impedance due to the presence of DSP, anti-CRP and CRP on nanofiber-coated electrodes.

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
