# Peer review of "Detection of Cardiovascular CRP Protein Biomarker Using a Novel Nanofibrous Substrate"

_biosensors, 2020, doi:10.3390/bios10060072_

Round 1

Reviewer 1 Report

This study reports the fabrication of an interdigitated sensing device integrated with polyvinyl alcohol (PVA) nanofibers and carbon nanotubes (CNT) for the detection of an inflammatory biomarker, C – reactive protein (CRP).

I believe this work is deserved to be reported by the journal, which may arouse great interest for relevant materials scientists to apply it. There are, however, several corrections needed and some changes I would suggest before acceptance:

  1. The abstract is very good written, however, introduction needs to be improved a bit with more details about the nanofibers with adding the previous efforts for the development of the biosensor based on nanofibers such as ACS Applied Materials & Interfaces 12 (2020), 6159−6168 and Biosensors and Bioelectronics 117 (2018) 838–844. As the nanofibers are considered a core of this study.
  2. What the thickness of the nanofiber membrane on the surface of the electrode, and how the authors controlled or optimize the electrospun polymer layer.
  3. Did the author study the effect of the thickness on the sensor performance, as the nanofiber layer thickness should have obviously influence the impedance results?
  4. Figure 1 caption: SEM is more precise to be Scanning Electron Microscope. Also, in Figure 1d, it will be better to add an SEM image with higher magnification to show the porous structure of the nanofibrous membrane.
  5. Figure 2 caption is error, it needs to revise. As it’s same with Fig 1 caption
  6. The authors should add XRD for produced PVA-CNT composite to prove the changes and common peaks in the product.
  7. Figure 3, the quality of the figure needs to improve. As the “X” axes title for Figure 3a should be added and the “X” axes title for Figure 3b should be inside the box
  8. Figure 5. the quality of the figure needs to improve.
  9. Figure 6. the quality of the figure needs to improve; add the axes titles and use lower case letters to match with the figure caption.
  10. In the manuscript, the author explained the excellent sensitivity performances of obtained nanofibrous membranes benefiting from large surface areas open porous structure. The large surface area requires specific data support, for the surface areas and pore size test or others.
  11. I suggest comparing the performance of the PVA-CNT nanofibers-based biosensor with the PVA-CNT casted membrane to show the advantages of assembling the sensor matrix in the nanofibers form.

Author Response

  1. The abstract is very good written, however, introduction needs to be improved a bit with more details about the nanofibers with adding the previous efforts for the development of the biosensor based on nanofibers such as ACS Applied Materials & Interfaces 12 (2020), 6159−6168 and Biosensors and Bioelectronics 117 (2018) 838–844. As the nanofibers are considered a core of this study.

We thank the reviewer for pointing these two articles out. Based on the reviewer’s suggestion, we have included the two articles on citric – acid decorated nylon nanofibers and PVA – co – PE nanofibrous membrane in the ‘Introduction’ section, page 2, lines 56 – 59 and added the two references [14] and [15].

  1. What the thickness of the nanofiber membrane on the surface of the electrode, and how the authors controlled or optimize the electrospun polymer layer?

That is an excellent question and we thank the reviewer for raising it. The thickness of the nanofiber substrate was controlled by the flow – rate of the syringe pump (determining the amount of PVA dispensed) and the amount of time that the nanofibers were allowed to be collected for. Based on the above parameters, the thickness of ~0.5mm was adequate enough to be tested for the determination of CRP. As the purpose of this study was to test the nature of the PVA/ CNT nanocomposite and to determine its ability to detect CRP, the optimization based on its thickness was not conducted. In future, we would certainly like to test the substrate’s ability to detect biological macromolecules based on variation of its thickness. We have included these details within the ‘Materials & Methods’ section on page 5, lines 138 – 140.

  1. Did the author study the effect of the thickness on the sensor performance, as the nanofiber layer thickness should have obviously influence the impedance results?

As we mentioned in comment 2 above, since we did not choose thickness of the substrate as one of the parameters to quantify the biosensor performance for, thickness variation was not studied. However, we do agree with the reviewer’s perception of the fact that changes in the thickness of the substrate should influence the impedance results to some extent.

  1. Figure 1 caption: SEM is more precise to be Scanning Electron Microscope. Also, in Figure 1d, it will be better to add an SEM image with higher magnification to show the porous structure of the nanofibrous membrane.

We thank the reviewer for pointing this out. We have changed the word ‘micrograph’ to ‘microscope’ in Figure 1 caption, page 4, line 118 and fixed the caption for Figure 1c on line 119. Also, as per the reviewer’s suggestion, we have replaced Figure 1D with a high resolution SEM image showing the porosity of the synthesized nanofibers on page 4.

  1. Figure 2 caption is error, it needs to revise. As it’s same with Fig 1 caption.

We thank the reviewer to point this error out. We have corrected the Figure 2 caption on page 8, lines 238 – 246.

  1. The authors should add XRD for produced PVA-CNT composite to prove the changes and common peaks in the product.

We thank the reviewer for the suggestion. However, at this time we would like to politely decline doing the XRD on the PVA/ CNT nanocomposite owing to the time and resources constraints.

  1. Figure 3, the quality of the figure needs to improve. As the “X” axes title for Figure 3a should be added and the “X” axes title for Figure 3b should be inside the box.

We thank the reviewer for helpful suggestion to improve the quality of the figure. As per the suggestion, we have added the X – axis label to Figure 3a and moved the X – axis label for Figure 3b inside the box on page 10.

  1. Figure 5. the quality of the figure needs to improve.

We thank the reviewer for their comment on Figure 5 regarding its quality. We have regenerated the data plots and added the error bars to improve the quality of the figure on page 14.

  1. Figure 6. the quality of the figure needs to improve; add the axes titles and use lower case letters to match with the figure caption.

We thank the reviewer for raising this point. We have added the axes titles and used the lower case letters for the figure captions on page 15 of the revised manuscript.

  1. In the manuscript, the author explained the excellent sensitivity performances of obtained nanofibrous membranes benefiting from large surface areas open porous structure. The large surface area requires specific data support, for the surface areas and pore size test or others.

We thank the reviewer for this comment on size of the pores. The only data that we have for estimating the pore sizes are the SEM images. As performing the porosity measurements would require additional resources and time, we used the SEM images to estimate the pore sizes which fall in the range of 100 to 500 square – nanometers. This information is included in the manuscript on page 5, lines 123 – 124.

  1. I suggest comparing the performance of the PVA-CNT nanofibers-based biosensor with the PVA-CNT casted membrane to show the advantages of assembling the sensor matrix in the nanofibers form.

We thank the reviewer for this suggestion. Indeed we used electrospun membrane. Individual fibers may have been laid on the electrode but still they are multiple fibers so in essence we are using membrane like matrix but taking advantage of higher specific surface area (SSA). We have added this information in the revised manuscript on page 2, lines 67 – 69.

Reviewer 2 Report

Report – Manuscript ID - biosensors-838673

The manuscript reports a novel biosensor for CRP detection based on electrochemical detection. The main proof of concept relies on a modified working electrode with PVA and carbon nanotubes. The biosensor presents an impressive LOD, although it lacks other performance parameters. Human samples were used in the biosensor proof of concept. I recommend the manuscript for publication in Biosensors, with minor corrections.

  1. The manuscript is too long, and it becomes difficult to read. I suggest the authors take the theoretical part that is in the results section.
  2. Recommend joining results and discussion section
  3. I recommend a diagram with the experimental setup showing the electrodes and fluid handling.
  4. Is it is not clear why DSP linker choice for antibody immobilization? This information should be added to the manuscript.
  5. Figure 3a – the steps in the figure were done using the same sensor surface? It’s confusing… this means that the sensor needs to regenerate, you need to remove all the protein from one step to the other with the washings… is this possible? Please make it clear in the manuscript either in figure capture or in text.
  6. How was LOD determined? There are many ways of calculating, it’s important that the reader knows which one you used.
  7. The slope of the curves is not very pronounced. What is the signal to noise ratio for the 1fg/ml and for the other protein concentrations? Is it easy to distinguish the signal between one concentration of protein and others?
  8. Figure 4 – difficult to understand why is this the biosensor performance. Performance parameters: precision, accuracy, dynamic range, sensitivity, specificity, and precision and dynamic range are not really mentioned. This is important.
  9. Explain the difference of response curves in serum and in PBS. If not sure, make a hypothesis with literature backup.
  10. The percentage change in impedance ranged from 45% to 70% (hs) and 38% to 60% (PBS) over a concentration range of 1 fg ml-1 to 100 ng ml-1. But does this difference translate into specific signal? What is the slope in the dynamic range, the range of CRP that can actually be quantified?
  11. How was the fluid handling made? How were the washings made? This information should be added.
  12. Figure 4 and 5 – please add error bars

Author Response

  1. The manuscript is too long, and it becomes difficult to read. I suggest the authors take the theoretical part that is in the results section.

We thank the reviewer on the suggestion based on which we have removed some part of the theory in the Results section on pages 7 and 8, lines 220 – 229 and 247 – 249.

  1. Recommend joining results and discussion section.

We thank the reviewer for this suggestion. However, as per the template of the Biosensors journal, all the authors agreed on keeping the Results and the Discussion sections separate so as to avoid any unnecessary confusion on the reader’s part. 

  1. I recommend a diagram with the experimental setup showing the electrodes and fluid handling.

As per the recommendation of the reviewer, we have added Figure 2c on page 8 on the experimental setup of the electrodes for the EIS measurement with a caption on lines 245 – 246.

  1. Is it is not clear why DSP linker choice for antibody immobilization? This information should be added to the manuscript.

We thank the reviewer for this comment. It is known that DSP, which is a covalent linker, has a thiol (sulphur) group at one end, which has a high affinity for the gold surface while the other end, which is the NHS group binds with the biological macromolecule (Anti – CRP in this case). Based on the highly porous structure of the nanofibrous substrate, the DSP linker permeated through these pores and anchored on the underlying gold substrate. This unique substrate now allows the interaction of the analyte (CRP) with the anti – CRP within these pores increasing the overall sensitivity of the biosensor. As per the reviewer’s suggestion, we have added this piece of information in the revised manuscript on page 6, lines 169 – 175. We have also added two references on this information, [35] and [36].

  1. Figure 3a – the steps in the figure were done using the same sensor surface? It’s confusing… this means that the sensor needs to regenerate, you need to remove all the protein from one step to the other with the washings… is this possible? Please make it clear in the manuscript either in figure capture or in text.

We thank the reviewer to point out this confusion. We performed the experiments for different concentrations on individual electrodes (sensor platforms) to avoid the residual protein removal and regeneration issues. We have clarified this in the revised manuscript, both in the Figure 3a caption on page 10, lines 267 – 268 and in the ‘Results’ section 3.1 page 11, lines 272 – 273.

  1. How was LOD determined? There are many ways of calculating, it’s important that the reader knows which one you used.

As indicated on page 11, lines 283 – 285 of the manuscript, the limit of detection was estimated based on the percent change in impedance from the control sample (antibody baseline impedance), which in this case was the anti – CRP mounted on the PVA/ CNT coated electrode. We have clarified this in the revised manuscript.

  1. The slope of the curves is not very pronounced. What is the signal to noise ratio for the 1fg/ml and for the other protein concentrations? Is it easy to distinguish the signal between one concentration of protein and others?

We thank the reviewer for clarification on this because as can be seen from Figure 3b, for concentration upto ~10pg/ml, we did find relatively larger differences in the impedance signals to distinguish between the different concentrations from 10pg/ml downto 1fg/ml. However, above 10pg/ml we did find that the biosensor saturated (limit of linearity from the performance standpoint) as mentioned on page 11, lines 293 – 294.

  1. Figure 4 – difficult to understand why is this the biosensor performance. Performance parameters: precision, accuracy, dynamic range, sensitivity, specificity, and precision and dynamic range are not really mentioned. This is important.

We gratefully acknowledge this concern of the reviewer on the caption of Figure 4. In fact, Figure 4 is the dose response of CRP and not the biosensor performance per say involving the dynamics. This is as explained on page 11, lines 285 – 293. We have rectified this in the caption of Figure 4 on page 13, line 298.

  1. Explain the difference of response curves in serum and in PBS. If not sure, make a hypothesis with literature backup.

Based on the suggestion of the reviewer, we have modified the following piece of discussion on the difference between the response curves in serum and in PBS on page 16, lines 349 – 353:

“The changes in the impedance were comparable in both the ionic buffer as well as in human serum. The minimization of high background in the human serum buffer may be attributed to the sieve like behavior of the nanofibers that provide size matched confinement of the target proteins. Furthermore, given the nature of the substrate being nanofibrous and porous, it turns out to be a better microenvironment for the analyte to interact. Also, the pores within the nanofibrous substrate would provide an environment that may promote phenomenon of macromolecular crowding that typically exists in confined spaces, thereby altering the behavior of human serum.”

  1. The percentage change in impedance ranged from 45% to 70% (hs) and 38% to 60% (PBS) over a concentration range of 1 fg ml-1 to 100 ng ml-1. But does this difference translate into specific signal? What is the slope in the dynamic range, the range of CRP that can actually be quantified?

This is a good comment. Based on our analysis, as can be seen from Figure 3, the differences between human serum and PBS are with respect to the obtained impedance values. It is from comparing these real impedance signal values to the anti – CRP baseline impedance value (control) that the percentage change in impedance as mentioned for hs and PBS were calculated. The change in this impedance when detecting the CRP is the signal that can be used to quantify any unknown concentration of CRP. Based on the known concentrations of CRP, as can be seen from Figures 3 and 4, the designed biosensor can detect CRP in the concentration range of 1 fg/ml to 100 ng/ml.

  1. How was the fluid handling made? How were the washings made? This information should be added.

We thank the reviewer for pointing this out. We have added the information on the PBS washings on page 6, lines 180 and 183 of the revised manuscript.

  1. Figure 4 and 5 – please add error bars.

We thank the reviewer for pointing this out. We have added the error bars to figure 4 and we have regenerated the graph for Figure 5 containing the error bars.